



# Long-term changes in the dependence of NmF2 on solar flux at Juliusruh

Maria Gloria Tan Jun Rios[1,2], Claudia Borries[1], Huixin Liu[2], and Jens Mielich[3]

[1]German Aerospace Center, Insitute for Solar-Terrestrial Physics, Neustrelitz, Germany
[2]Department of Earth and Planetary Science, Kyushu University, Japan
[3]Leibniz-Institute of Atmospheric Physics, Rostock, Germany

**Correspondence:** Maria Gloria Tan Jun Rios (maria.tan@dlr.de) and Huixin Liu (liu.huixin.295@m.kyushu-u.ac.jp)

**Abstract.** Understanding the ionospheric dependence on solar activity is crucial for the comprehension of the upper atmosphere. The response of the ionosphere to solar EUV flux has been previously considered stable. Subsequent studies have revealed long-term changes that are not yet fully understood. This work evaluates the stability of the $NmF2$ dependence on solar EUV indices throughout different solar cycles.

Hourly values of the peak electron density of the ionospheric F2-layer ($NmF2$) from Juliusruh station (54.6°N, 13.4°E) are analysed between 1957 and 2023. Geomagnetic perturbations are removed. Third-degree polynomial fit models dependent on different solar EUV proxies (MgII, F30, and F10.7) are generated for each solar cycle and for each season separately.

The saturation effect is visible in our data and starts at lower F30 values in the ascending phase than in the descending phase. A well-pronounced local time dependence in January with the $R^2$ value being maximum around noon hours has been observed.

The correlation is highest for F30 and MgII especially during winter noon conditions, supporting recent studies that they are the best solar flux proxies for describing the $NmF2$ dependence at all LT hours. Most importantly, the response of $NmF2$ to solar flux shows a clear long-term change as the slope of the model curves decreases with time for each solar cycle. Between SC20 and SC24, the observed decrease is consistently higher than 16%, reaching 24% at 90 sfu which means a decal reduction of 3-4.4% between 1954 and 2019.

## 1   Introduction

The ionospheric variations over time are an important part of space climate because they can change ionospheric conditions for HF radio communications and propagation.

Investigations of long-term changes in the upper atmosphere and ionosphere began with the pioneering study of Roble and Dickinson (1989). They suggested that "greenhouse cooling" (Cicerone, 1990) could occur in the upper atmosphere due

to the long-term increase in greenhouse gas concentrations ($CO_2$) with the subsequent atmospheric contraction leading to a lowering of ionospheric layers. Modeling studies by Rishbeth (1990) and Rishbeth and Roble (1992) broadened these results to the thermosphere–ionosphere system. With the increasing number of observational and model results and findings, a global pattern of trend behavior was constructed (Laštovička et al., 2006, 2008). After that, other parameters appeared in this scenario playing an important role in long-term trends in the upper atmosphere and ionosphere together with the dominant increase of



atmospheric concentration of greenhouse gases, for instance, secular change of Earth's magnetic field, changes in stratospheric ozone and others.

The ionosphere is mainly formed due to the ionizing effect of solar extreme ultraviolet (EUV) radiation. Changes in solar activity impact the chemical reactions and physical processes within the system. The main driver of year-to-year changes in ionospheric characteristics is the quasi-eleven-year solar cycle and for that reason, understanding the solar activity dependence of the ionosphere is crucial for empirical models.

Solar EUV is mostly absorbed before reaching the lower atmosphere, which makes accurate ground-based monitoring challenging. Space-based measurements have been intermittent. As a result, solar EUV proxies have been used to model solar EUV emissions due to the lack of long-term records (e.g., Richards et al., 1994; Mikhailov and Schlegel, 2000). The question of which solar activity proxy is the best representation of EUV is still open, but numerous recent studies indicate that F30 and MgII are the most reliable proxies for long-term analysis (Laštovička, 2021; Danilov and Berbeneva, 2023; Laštovička and Burešová, 2023; Zossi et al., 2023).

The global network of ionosondes provides the critical frequency of the F2-layer, $foF2$, with very long data series in some stations. As changes in the peak electron density of the F2-layer, $NmF2$, directly quantify changes in the F2-layer ionosphere, it is an ideal parameter to be used for analyzing long-term trends in the ionosphere. $NmF2$ can be easily derived from $foF2$ data.

The $NmF2$ response to solar EUV proxies (F10.7 or R) was found to be linear in early studies (Bremer, 1992; Laštovička, 2024), and it is often used for ionosphere analyses and modeling. However, later studies (e.g., Liu et al., 2003; Chen et al., 2008; Balan et al., 1994, 1996) discovered that the linear increase of $NmF2$ with solar EUV proxies at low and moderate solar activity levels breaks down at higher activity levels, indicating a "saturation effect" and consequently, a nonlinear dependence (Balan et al., 1994). Recent publications (e.g., Liu et al., 2006; Ma et al., 2009; Danilov and Berbeneva, 2023) show that the dependence of $foF2$ to solar flux is better represented with a third-degree polynomial regression and its dependence on diurnal and seasonal variation. Kouris et al. (1998) found that using a higher-order polynomial did not effectively improve the fitting.

It is still an open debate if the ionospheric saturation effect is a genuine manifestation of solar activity and the root cause of this effect. Balan et al. (1994) suggested that the ionospheric saturation effect is due to the nonlinearity of EUV radiation with solar EUV proxies. However, Liu et al. (2003) found that the ionospheric saturation effect still appears for EUV radiation measurements depending on the geographical location, revealing that the nonlinearity cannot fully explain the saturation effect. Rather, the roles of ionospheric photochemistry, the neutral upper atmosphere, and dynamics also contribute to the solar activity changes in $NmF2$ (Liu et al., 2006). In addition, ionospheric characteristics may have different values for the same solar level during different phases of a solar cycle, which is known as the 'hysteresis' effect (e.g., Mikhailov and Mikhailov, 1995; Rao and Rao, 1969; Triskova and Chum, 1996).

The manuscript's working hypothesis is that long-term changes in ionospheric characteristics reported previously should be evident in their response to solar activity. This work studies and quantifies the long-term change in the ionosphere at the mid-latitude station Juliusruh (Germany) by parametrization of the ionospheric response to solar activity for each separate solar cycle. We use the existing knowledge of the most relevant solar activity proxies in long-term analysis (F30, MgII, and F10.7)



and their non-linear relation to ionospheric characteristics by utilizing a third-degree polynomial fitting. The relevance of the hysteresis effect will be studied by analyzing the ascending and descending phases of each solar cycle separately. $NmF2$ is the ionospheric parameter considered in this paper and for the sake of comparison with other works, the results concerning $foF2$ are provided in the Appendix 5. 5.

## 2  Data and methods

### 2.1  Solar Activity Index

To study the ionosphere and thermosphere the coorelation with solar extreme ultraviolet (EUV) radiation plays an important role. Ground-based equipment is not able to monitor EUV as it gets absorbed before entering the lower atmosphere. To measure solar EUV fluxes, rockets, satellites, and indirect methodologies have been used. However, direct (space-borne) measurements of the solar EUV spectrum and its variability are not available for most of the time. Consequently, scientists rely on solar EUV
proxies to indicate the intensity of solar activity.

Each solar proxy corresponds to different parts of the solar radiation spectrum; therefore it is possible to obtain different results using different solar activity proxies. Three solar activity proxies are used here: F10.7, F30 and MgII index. F10.7 and F30 are measures of the solar radio emission at a wavelength of 10.7 centimeters (2.8 GHz frequency) and at a wavelength of 30 centimeters (1 GHz frequency) respectively. The Mg II core-to-wing index originates from the chromosphere and is
computed by comparing the h and k lines of solar Mg II emission at 280 nm with the background solar continuum near 280 nm.

All the mentioned solar activity proxies have a daily resolution. To make an hourly analysis, we apply the daily value corresponding to a particular day to all the hours of this day

For further analysis, the period corresponding for the last seven solar cycles were defined using a 3-year moving window
average of F30 values. With this window average we could reduce the fluctuation of the daily data and determine maximums and minimums occurrence (see Table 1). In addition, Figure 1 illustrates the variation of F30 over the years, the 3-year moving window average and the solar cycles. The ascending and descending part of each cycle is displayed with colored spans, the blue background indicates the descending part.

### 2.2  Juliusruh ionosonde data

The critical frequency of the F2-layer, $foF2$ data of station Juliusruh (54.61°N, 13.41°E) is considered with an hourly resolution for the period of six solar cycles (1957–2024). Juliusruh is a recommendable ionosonde station for long-term studies because the length of the data, the minimum data gaps and the homogeneous and high-quality data (Laštovička et al., 2006).

The peak electron density in the F2 layer values, $NmF2$, derived from the $foF2$ data using the following relation (Piggott and Rawer, 1972):

$$NmF2 = 1.24 \times 10^{10} \cdot (foF2)^2 \tag{1}$$



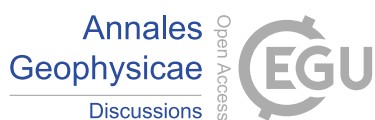

**Table 1.** Definition of solar cycle periods used in this work, date of the maximum solar cycle occurrence and the 3-year moving window average maximum F30 value and minimum F30 value.

| solar cycle (M.Y) | | Date of Max. (M.Y) | Max Av.F30 (sfu) | Min Av.F30 (sfu) |
|---|---|---|---|---|
| SC19 | 04.1954 to 06.1964 | 03.1958 | 142 | 46 |
| SC20 | 06.1964 to 12.1975 | 06.1969 | 96 | 48 |
| SC21 | 12.1975 to 03.1986 | 07.1980 | 125 | 49 |
| SC22 | 03.1986 to 02.1996 | 09.1990 | 130 | 50 |
| SC23 | 02.1996 to 06.2008 | 05.2001 | 120 | 46 |
| SC24 | 06.2008 to 04.2019 | 01.2014 | 93 | 46 |
| SC25 | from 04.2019 | | 105 (until now) | |

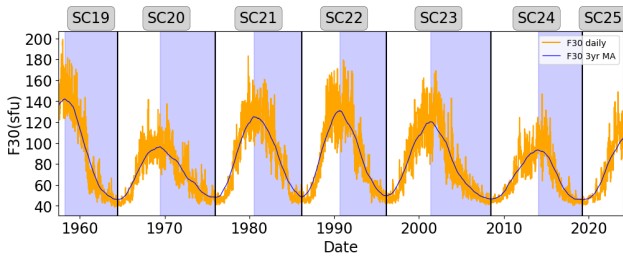

**Figure 1.** F30 (sfu) data from 1957 to 2024. Solid-line in blue indicates the 3-year moving window average of F30. The vertical black lines indicate the last solar cycles and the blue background indicates the descending part of each solar cycle.

where $NmF2$ is given in $m^{-3}$ and $foF2$ in $MHz$.

Figure 2 shows the Juliusruh ionosonde observations of $NmF2$ and some technical changes in this ionosonde which can affect the measurements. According to Sivakandan et al. (2023), since 1957 a high-power ionosonde has been working in Juliusruh. From 1990 to 1994 this ionosonde was replaced by a Polish ionosonde of type KOS and in 1994, it was replaced by a Digisonde (Reinisch et al., 1992, 2008). Additionally, from 1957 to 1993, the human scaling of the data was performed by different individuals, while since 1993, only one person has been responsible for this task.

### 2.3 Ionospheric data cleaning

For our analysis, we are using hourly data resolution from the ionosonde. To ensure that the data we are using is reliable and free of non-natural values or outliers caused by instrumental bias, we need to clean the data. We also want to exclude data from geomagnetical influenced days, as we are studying the behavior of the ionosphere during geomagnetic quiet days with Kp < 3. Our cleaning method involves two steps, which are outlined below.





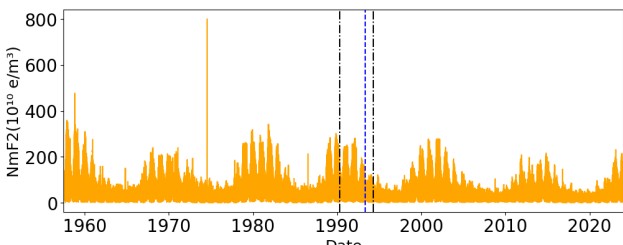

**Figure 2.** Juliusruh ionosonde hourly observations of $NmF2$. Vertical black dashed lines indicate a change in the ionosonde instrument and blue dashed line indicates the beginning of human scaling made for only one person.

**Table 2.** Quantified analysis of the $NmF2$ data and corresponding percentage for the cleaning method applied in this work.

| Total Values | Initial nan | Natural outliers | Geomagnetically influenced | Total after cleaning |
|---|---|---|---|---|
| 579575 | 48648 | 1740 | 149426 | 379761 |
| **100.0%** | **8.4%** | **0.3%** | **25.8%** | **65.5%** |

The first step entails the removal of all values that fall far outside the natural range of $NmF2$. These outliers are readily identifiable in Figure 2. Consequently, we deemed $NmF2$ values exceeding $4*10^{12}$ $e/m^3$ as outliers and removed them from our dataset.

To enhance our comprehension of the relationship between solar activity proxies and ionospheric characteristics, the second step consists of filtering our dataset to exclude geomagnetic disturbed days. Geomagnetic storms elicit an impact on ionospheric characteristics for on average two days for a moderate geomagnetic storm following their onset (Yokoyama and Kamide, 1997), and it can be even more, for instance, St. Patrick's Day storm in 2015 (Astafyeva et al., 2015). Consequently, to eliminate such disturbed periods, we have identified and removed days where the Kp index is equal to or exceeded 3, as well as the 48 hours succeeding them.

Table 2 displays the amount of $NmF2$ data that resulted after implementing a cleaning method explained above. The first column indicates the total number of data points selected for the period, the second column shows the number of initially missing values (not a number, nan) from the ionosonde data. The third column indicates the values that are far outside the natural range of values (first step). The fourth column shows the removed data to correspond to the disturbance geomagnetic days. Finally, the last column shows the amount of data remaining after the cleaning method used. All columns show the actual number of points and the percentage they represent regarding the total number of values shown in the first column.





## 2.4 Methodology: Regression analysis

The ionospheric response to the solar flux is represented using polynomial fitting, which is preferred over other methods. A cubic fitting for each month for each local time is used in the regression analysis to study the nonlinear correlation between

$NmF2$ (variable Y) and the solar EUV proxy (variable X).

The equation 2 shows the regression used.

$$Y = a_0 + a_1 X + a_2 X^2 + a_3 X^3 \tag{2}$$

The data was grouped according to the solar cycle, and the goodness of the description for each fit is indicated with the $R^2$ value. $R$, indicted in equation (3), is the correlation coefficient between the time series:

$$R = \sum_{i=1}^{N} (x_i - \bar{x})(y_i - \bar{y}) \Big/ \sqrt{\left( \sum_{i=1}^{N} (x_i - \bar{x})^2 \right) \left( \sum_{i=1}^{N} (y_i - \bar{y})^2 \right)} \tag{3}$$

Figure 3 displays an example of the linear and polynomial fit using $NmF2$ data and solar EUV proxy F30 for January at 14 LT during solar cycle 22. The data is represented in blue scattered points, the red solid line represents the standard third-degree polynomial fit, and the green solid line indicates the standard linear fit resulting from the data. Robust regression methods, which use iteratively reweighted least squares to assign a weight to each data point and are less sensitive to outliers than

standard regressions, were tested and brought the same results, indicating that there are no significant outliers in the data and consolidating the data cleaning method explained in section 2.3.

Additionally, the confidence interval of the polynomial fit between the ionospheric parameter and the solar EUV proxy for a particular LT in all Januaries of each solar cycle was calculated using the Bootstrap method. In Figure 3, the confidence interval is shown as a translucent stripe around the polynomial fitting line. The bootstrap method computes confidence intervals

without relying on the assumptions of standard theory, making it useful for both parametric and non-parametric applications. The process involves re-sampling with replacement from the original dataset to create a new dataset. More information about this method can be found in the bibliography (e.g. Hall, 1992; Efron and Tibshirani, 1994; Mansyur and Simamora, 2022).

To support our findings and assess the quality of the fits, we employ an alternative methodology. This involves clustering the data based on a specific local time (LT) hour, month, and solar cycle, similar to the previous method. We then create a

histogram with 20 bins for each cluster. For every bin in the histogram, we calculate the mean value and consider the standard deviation as an error of this value. Additionally, we identify the bins with fewer than 10 values to show that these bins carry less weight in our results. In Figure 3 the black scatter points represent the mean values, while the crosses correspond to the mean values in the bin with less than 10 counts.

This bins approach is unbiased when it comes to statistical fittings, and it supports the findings obtained through the poly-

nomial fitting methodology.

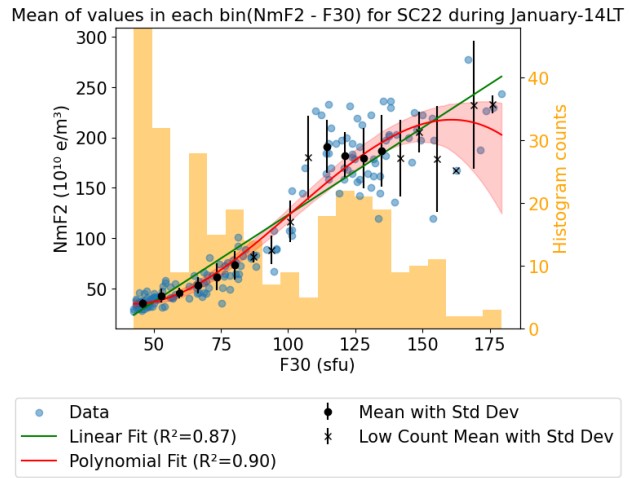

**Figure 3.** Linear (green line) and polynomial (red line) fits dependence between $NmF2$ and F30 during January at 14 LT during solar cycles 22. Mean values of the bins (black scatter points) and mean values with less than 10 counts in the bin (cruises) with their standard deviation (error bar for each point).

## 3 Results

### 3.1 Seasonal Analysis with different solar EUV proxies

This study investigates the seasonal influence on the variation of $NmF2$ solar activity by analyzing the $R^2$ value obtained in the third-degree polynomial fit between $NmF2$ and the solar EUV proxies. The analysis is done on an hourly fitting for each month from 1957 to 2023. Figure 4 shows four months, one for each season: January for winter, April for spring, July for summer, and October for fall.

From Figure 4, it is apparent that during January (winter), there is a clear diurnal variation in the $R^2$ values. This means that the correlation increases abruptly during morning hours, reaching a maximum $R^2$ value of 0.85 at 12-17 LT, and then decays in the evening. Furthermore, this season exhibits the highest variability between night and noon time compared to the rest, with $R^2$ values ranging from 0.1 at 4 LT to 0.85 at 12-17 LT. On the other hand, during April (spring) and July (summer), the diurnal variation is not visible, as $R^2$ remains constant between 0.6-0.8. In October (fall), the diurnal variation is visible, but with less variability of $R^2$ values between night and noon time than in winter.

Finally, it is observed that the highest correlations between $NmF2$ and solar EUV proxies for each hour at different months (1957-2023) are always reached using F30. Using MgII, the $R^2$ values are almost equal to those of F30 in January and October. In April and July, the correlations using MgII and F10.7 are mostly overlapping each-other and lower than those with F30. The red line, corresponding to F10.7, shows the lowest values of $R^2$ in all cases. $R^2$ values do not differ significantly when using either a linear or a polynomial regression (See Figure B1 in the appendix). The highest correlations over time during winter noon hours, allow us to continue the long-term analyses under this condition.

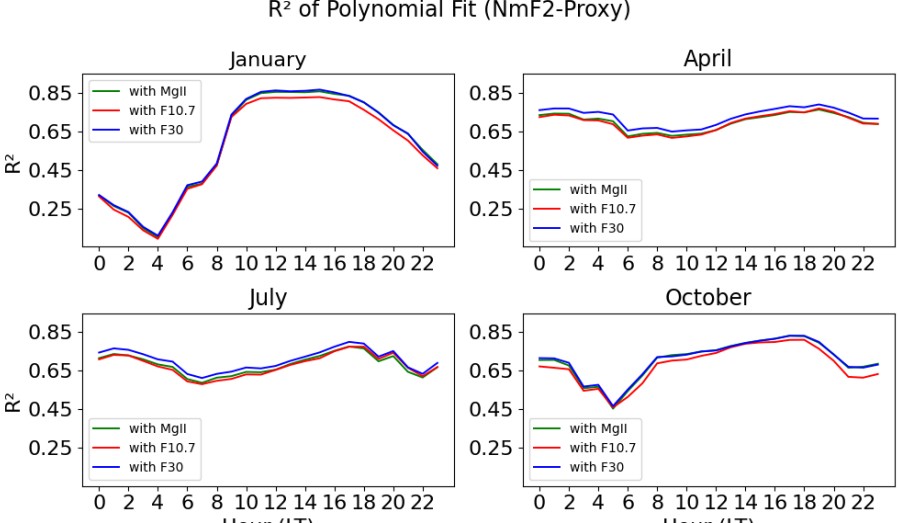

**Figure 4.** Hourly $R^2$ value of the third-degree polynomial dependence between $NmF2$ and solar activity proxies: F30 (blue line), F10.7 (red line) and MgII (green line); a)in January; b)in April; c)in July and d)in October from 1957 to 2023.

## 3.2 Long-term changes

This work aims to investigate the continuity of the relationship between $NmF2$ and solar flux across different solar cycles. To achieve this, we divide the period from 1957 to 2023 into different solar cycles based on Table 1 and consider the period between solar cycles 20 to 24 when the observations are available for each full solar cycle. We analyse the ionospheric response to solar activity proxies, represented by the third-degree polynomial fit. We utilize data from January during noon hours and the solar EUV proxy F30 because the results in 3.1 reveal the highest correlation under these conditions.

In Figure 5, the curves representing the polynomial fit for each solar cycle appear to have a systematic shift based on the solar cycle number. In other words, for a specific solar flux levels, the $NmF2$ values decrease with increasing solar cycle number. This is most noticeable for lower F30 values where the curves have similar slopes. For higher F30 values, the saturation effect makes the ordered shift of the curves less apparent. The saturation effect is observable in the first three solar cycles (SC20 to SC22) but is not evident in SC24. In SC23, the saturation effect is absent, and the ionospheric response increases more

significantly than before at higher solar flux levels.

To assess the significance of the differences between SC20 and SC24, the left panel of Figure 6 displays the polynomial fit between $NmF2$ and F30 in January at 14 LT of the first solar cycle (SC 20) and the last one analysed (SC 24), along with their corresponding 95% confidence intervals calculated using the Bootstrap method (explained in Sec. 2.4). The figure includes the methodology with the mean values of the histogram bins and their standard deviation error for each solar cycle. There are few

overlaps between the data points of SC20 and SC24 for up to 60 sfu and higher $NmF2$ values in SC20. Above 70 sfu, the





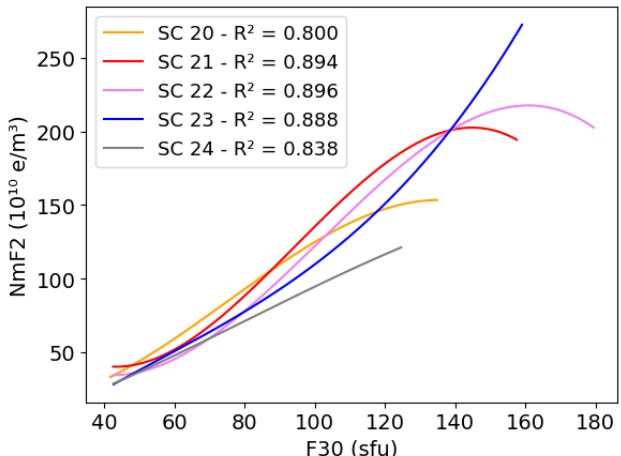

**Figure 5.** Third-degree polynomial dependence between $NmF2$ and F30 during January at 14 LT for different solar cycles.

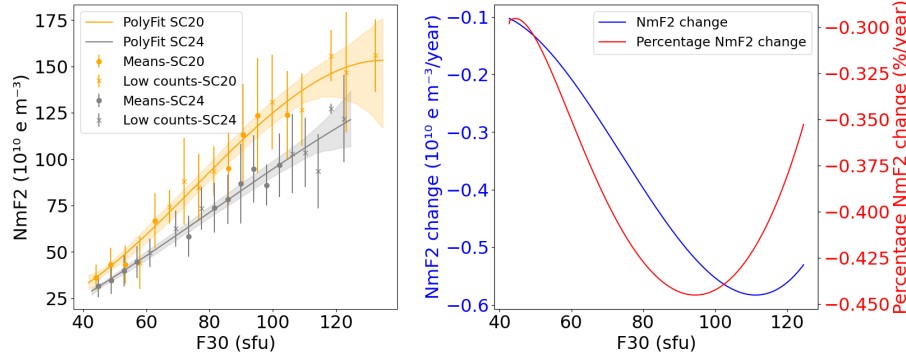

**Figure 6. Left panel**: Third-degree polynomial dependence between $NmF2$ and F30 during January at 14 LT for solar cycles 20 and 24 with their confidence intervals (CI) indicated as shades of the same regression line color. Mean values of the bins (scatter points) and mean values with less than 10 counts in the bin (crosses) with their standard deviation (error bar for each point); **Right panel**: Absolute and percentage per year differences between the third-degree polynomial fitting corresponding to solar cycles 20 and 24.

difference between the two solar cycles becomes more obvious, indicating a decrease in the ionospheric response to solar flux between the first (SC20) and the last part of the analyzed period (SC24).

The right panel of Figure 6 shows the differences between the two polynomial fits of SC 20 and SC 24. The higher the solar flux, the greater the decrease in the response over time. For 60 sfu, the difference between SC20 and SC24 is around $2*10^9$
$e/m^3$ per year, while for 120 sfu, the absolute variation is approximately $5.5*10^9$ $e/m^3$ per year. Additionally, it is worth noting that the observed decrease between SC20 and SC24 is consistently higher than 0.3% per year, reaching 0.44% per year at 90 sfu, that is a decal reduction of 3-4.4%. Between SC20 and SC24 we observed a decrease of 16% to 24%, what is a 3.2%-4.8% per solar cycle.





### 3.2.1 Ascending/descending phases of the solar cycle

A similar process to the one described in 3.2 is carried out here, but now only analyzing data from the ascending (and descending) part of all solar cycles separately. Figure 1 describes the solar cycles and their phases used here. Table 1 indicates the beginning and end of each solar cycle and also their maximum, which indicates the end of ascending and the start of descending phase.

The variability in the response of the ionosphere to the solar EUV proxy (F30) over time in the ascending phases of solar
cycles is shown in Figure 7. Moreover, Figure 8 presents the differences in the fitted response in the first and last solar cycles analysed including the significance of these (95% confidence interval). The first solar cycle analysed here is SC20, and the last solar cycle analysed is SC25 because we considered that the data until December 2023 more or less includes a big portion of the present solar cycle. Likewise, the order shift mentioned in Figure 5 is not as clear here and the difference between the curve in SC20 and SC25 is not significant at the lowest values of $NmF2$. The confidence intervals for the polynomial fitting of the
ascending phases in SC20 and SC25 partially overlap for all $NmF2$ values, indicating the possibility of a small difference in the response over time. For 60 sfu, the difference between SC20 and SC25 is around $2*10^9$ $e/m^3$ per ascending year, while for 120 sfu, the absolute variation is approximately $7.5*10^9$ $e/m^3$ per ascending year. The observed decrease between SC20 and SC25 is consistently higher than 0.4% per ascending year, reaching 0.53% per ascending year at 100 sfu.

Figure 9 and 10 illustrate the descending phase of the solar cycles. The analysis includes data from SC19 through SC24.
The shift in order is more noticeable than in Figure 5 for the first solar cycles. The difference between the curves representing SC19 and SC24 in Figure 10 clearly shows that SC19 is stronger than SC24. Around 130 sfu, the polynomial fitting does not accurately represent the mean of that particular bin. However, this mean is calculated with fewer than 10 data points, making it less reliable. For 60 sfu, the difference between SC19 and SC24 is around $4*10^9$ $e/m^3$ per descending year, while for 120 sfu, the absolute variation is approximately $1.5*10^{10}$ $e/m^3$ per descending year. The observed decrease between SC19 and SC24
is consistently higher than 0.65% per descending year, reaching 0.9% per descending year around 100 sfu.

### 4 Discussion

The varying intensities of the solar cycle are a challenging point for our analysis. The difference in the strength of solar cycles results in different ionospheric responses, making some comparisons less straightforward. For instance, SC19 has a significantly larger amplitude than SC24, as shown in Figure 1. It could be argued that the difference between them is not due
to long-term changes but rather to a solar phenomenon. However, it is also evident that the ionospheric response decreases over time during periods of low solar activity, indicating a long-term change.

In order to study the long-term changes in the ionosphere, we choose periods when the EUV variability dominates the ionosphere variability. Jakowski et al. (2024) studied the long-term behavior of production and loss coefficients. They discuss that photoionization depends on the incidence angle of solar radiation. In summer, plasma transport dominates over recombination
processes and the peak electron density can occur after sunset, creating the Midlatitude Summer Nighttime Anomaly (MSNA). In winter, the peak electron density decreases around sunset, suggesting that recombination processes dominate at that time.





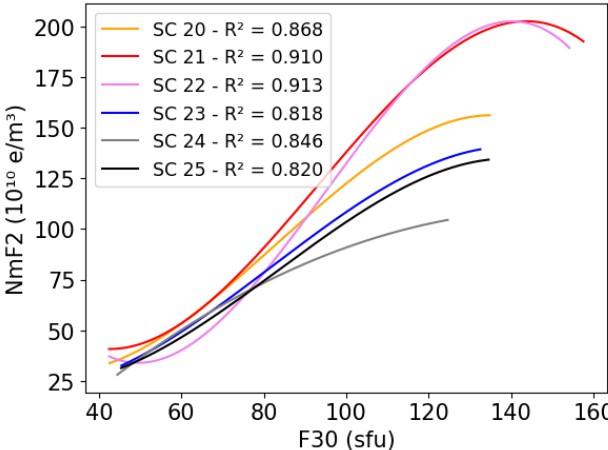

**Figure 7.** Third-degree polynomial dependence between $NmF2$ and F30 during January at 14 LT for the different Ascending part of each solar cycles.

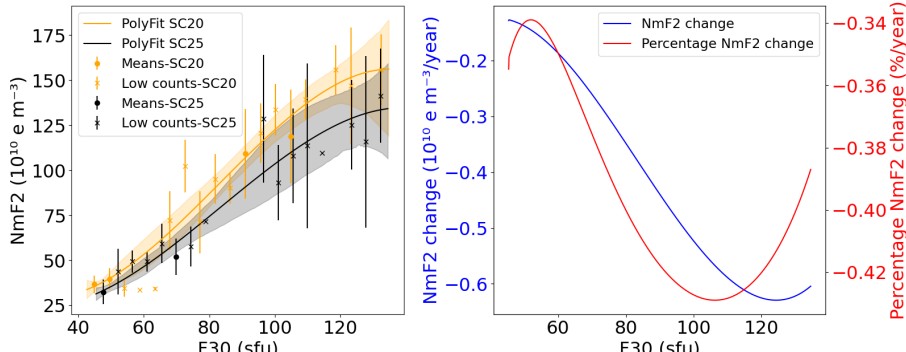

**Figure 8. Left panel**: Third-degree polynomial dependence between $NmF2$ and F30 during January at 14 LT for ascending phase of solar cycles 20 and 25 with their confidence intervals (CI) indicated as shades of the same regression line color. Mean values of the bins (scatter points) and mean values with less than 10 counts in the bin (crosses) with their standard deviation (error bar for each point); **Right panel**: Absolute and percentage differences per ascending year between the third-degree polynomial fitting corresponding to ascending phase of solar cycles 20 and 25.

In addition, there is a return flux of plasma from the plasmasphere in winter contributing to an increase of ionization and the Nighttime Winter Anomaly (NWA) in the Northern hemisphere (Jakowski and Paasch, 1984; Jakowski and Förster, 1995). Finally, winter conditions show the highest ratio of production and loss of ionization and the noon condition is the period with the strongest impact of solar ionizing flux. Accordingly, Figure 4 shows the highest correlations in January at noon hours, thus, the results discussed in the following use January 14 local time conditions only.





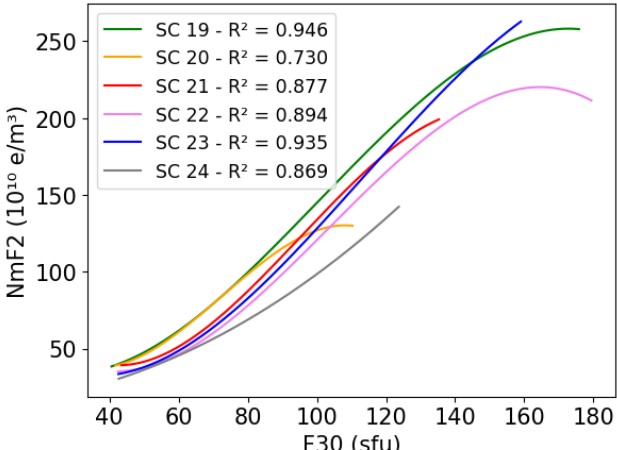

**Figure 9.** Third-degree polynomial dependence between $NmF2$ and F30 during January at 14 LT for the different descending part of each solar cycles.

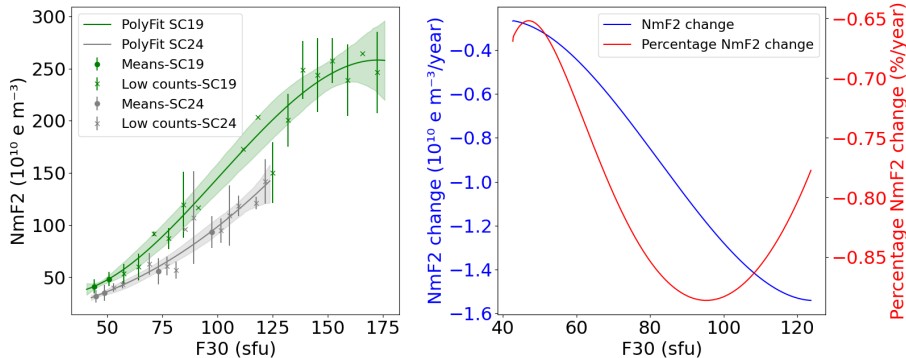

**Figure 10. Left panel**: Third-degree polynomial dependence between $NmF2$ and F30 during January at 14 LT for descending phase of solar cycles 19 and 24 with their confidence intervals (CI) indicated as shades of the same regression line color. Mean values of the bins (scatter points) and mean values with less than 10 counts in the bin (crosses) with their standard deviation (error bar for each point); **Right panel**: Absolute and percentage differences per descending year between the third-degree polynomial fitting corresponding to the descending phase of solar cycles 19 and 24.

In this work, we showed that the solar activity effect on the ionosphere can be accurately modeled using a third-degree polynomial fit which considers the saturation effect (Liu and Chen, 2009). So far, many studies of long-term changes prefer linear regression against polynomial fitting (e.g. Bremer, 1992; Laštovička, 2024). One reason might be that the monthly
median values used in these studies reduce the saturation effect and make the relationship to the ionizing flux more linear. However, there are also often arguments that linear regression coefficients are easier to interpret and that polynomial fitting




inherits the risk of overfitting. Thus, different measures have been taken in this work to argue the use of the third-degree polynomial fit (e.g. considering the mean values of data bins)

First, we analyse the character of the relation relation between $NmF2$ and the solar EUV proxy by analysing the fitting
coefficients (Eq. 3). Figure 12 shows the variation of the coefficients from the third-degree polynomial fitting over the different solar cycles in 5. The left column of Figure 13 shows, in different colors, the constant ($Y = a_0$), linear ($Y = a_1 X$), quadratic ($Y = a_2 X^2$), and cubic ($Y = a_3 X^3$) contributions in the polynomial fitting. The right column of this figure displays the percentage contribution of each part in the final polynomial fitting for each solar cycle. The linear contribution is higher for smaller solar EUV proxy levels and decreases for higher levels, showing the saturation effect. For most solar cycles, at higher EUV
proxy levels, the quadratic and cubic contributions dominate, with approximately 50% and 30%, respectively. SC23 exhibits a specific behavior, with the linear component dominating more than 40% until 125 sfu. At 125 sfu, the cubic and quadratic components start to increase, resulting in an equal contribution from all components at 160 sfu. These results support the use of a third-degree polynomial fit to model the ionospheric response to solar EUV flux because all components are not negligible. Second, we address the argument of overfitting by adding the average $NmF2$ per binned F30. These points are shown in Figs.
6, 8 and 10 to proof the correctness of the fits when it comes to the estimation of the long-term change. In Figure 5, SC22 shows an unusual behavior in the range of F30 between 40 and 70 sfu with much lower $NmF2$ than the other solar cycles. Overfitting would be a first guess to explain this unusually low $NmF2$. But, the bins analysis in Figure 3, which shows the $NmF2$ data for SC22 and its polynomial fit, indicates that the shape accurately aligns with the mean bins values of the data. Thus, it must be a natural effect causing the low $NmF2$ values and no artefact of the fitting function.
Finally, we complement the results using linear fitting in the appendix (Sec. B, Figs. B2, B3). They confirm the results discussed here using this alternative method.

The results of the polynomial fit for each solar cycle separately indicate that the relation between solar activity and $NmF2$ is not steady. The regression lines have similar slopes but shifted from one SC to the next. The shift is slow and comparing SC20 and SC24 a significant difference developed indicating a variation in the ionospheric response over time. SC23 does
not seem to follow the common curvature shape of the previous SCs. Figure 6 shows a percentage yearly decrease in the ionospheric response to F30 between SC20 and SC24 of 0.3-0.44% (or $2*10^9$-$5.5*10^9$ $e/m^3$/year) and a decadal reduction of 3-4.4%. This decrease is comparable with yearly decline of 0.15-0.23% (or 0.008-0.024 MHz/year) for $foF2$ between SC20 and SC24 in Figure A3, indicating a decadal reduction of 1.5-2.3%. This last result is consistent with the decadal trend of -1.8% reported in Table 2 of Laštovička (2024) for Juliusruh (1976-2014) even though our analysis covers a larger period. Our
findings also align with the results obtained using both standard and modified linear regression methods in the work of Cnossen and Franzke (2014), which showed a trend of -7.7 kHz per year for Juliusruh (1959-2005) as reported in Table 3 and Figure 5 of the mentioned paper. In both cases, the approach differs from ours but the results obtained are similar.

The ionospheric response to solar activity's decrease is not yet fully understood. Recent research results suggest follwing effects as main mechanisms causing the long-term changes in the ionospere: First, the dynamic effect of neutral winds and
electric fields on $NmF2$ modify the plasma transport on long time scales (Liu et al., 2006). Second, slow changes in the Earth's magnetic field and geomagnetic activity trends are able to produce some trends in F2-region and also to explain some





seasonal and daily variation patterns in trend values (Elias and de Adler, 2006; Cnossen and Richmond, 2013). In addition, changes in the composition of the thermosphere caused by the contraction of the atmosphere, such as the ratio of [O]/[N2], can have a significant impact on the ionosphere. The density and temperature of the neutral particles in the thermosphere increase

with higher solar activity due to the greater heating from solar UV radiation and ion drag (Guo et al., 2007).

The next discussion point addresses the analysis of the ascending and descending solar cycle phases. The variations in $NmF2$ during different solar cycle phases is knwon as hysteresis effect. This phenomenon is usually observed in $foF2$ data. It means that the same solar level can have different $NmF2$ values during different phases of a solar cycle. Mikhailov and Mikhailov (1995) suggested that the effect is related to differences in geomagnetic activity during the ascending and descending

phases (usually stronger during the descending phase). Nevertheless, we show here that the hysteresis effect is still visible in NmF2 and foF2 data using only Kp below 3 conditions. In Figure 7, when only the ascending phases are considered, the order shift mentioned for Figure 5 is not clear. SC21 and SC22 have higher values compared to the rest of the SCs. Moreover, the difference between SC20 and SC25 (Figure 8) varies between 0.4% and 0.53% per ascending year (between 12% and 15.5% over the ascending phases of the six solar cycles, that is 2-2.6% per solar cycle), however, the visible overlapping between

some parts of their confidence intervals could imply that this difference is not significant.

For the descending phases of the solar cycles, our results show a clearer order in the shift of the curves, especially for lower F30 values, i.e., 40-80 sfu (Figure 7). It is worth noting that in the descending phase, SC23 does not follow the common curvature shape of the previous SCs as in the full solar cycle analysis. Furthermore, the difference between SC19 and SC24 (Figure 10) varies between 0.65% and 0.9% per descending year showing a significant decrease in the ionospheric response

after five solar cycles (between 24% and 32% over the descending phases of the six solar cycles, that is 4.8-6.4% per solar cycle). Figure 11 compares the descending phase of SC19 and SC22, it could be more appropriate here due to the similar amplitude of these solar cycles. The difference between SC19 and SC22 indicates a decrease that varies between 0.5% and 1% per descending year (between 12% to 24% over the descending phases of three solar cycles, that is 4-8% per solar cycle), which seems to agree well with the 0.65-0.9% per descending year in Figure 10.

In summary, the results presented here indicate that a long-term change in the solar activity dependence is stronger and clearer to identify in the descending phases of the SCs. However, the differences between the two phases could have their origin in the diverse phenomena that occur during each phase.

A key result of our study is that the magnitude of the long-term decrease in NmF2 depends on the magnitude of the solar activity index. For small F30, the long-term change is approximately 3.2% per solar cycle. For F30 = 120 sfu, it is 4.8% per

295 solar cycle.

## 5 Summary and Conclusions

An analysis of hourly data on $NmF2$ derived from Juliusruh data, covering the period from 1957 to 2023, was conducted. The study examined the response of $NmF2$ to solar flux by using three different solar EUV proxies (F10.7, F30 and MgII).





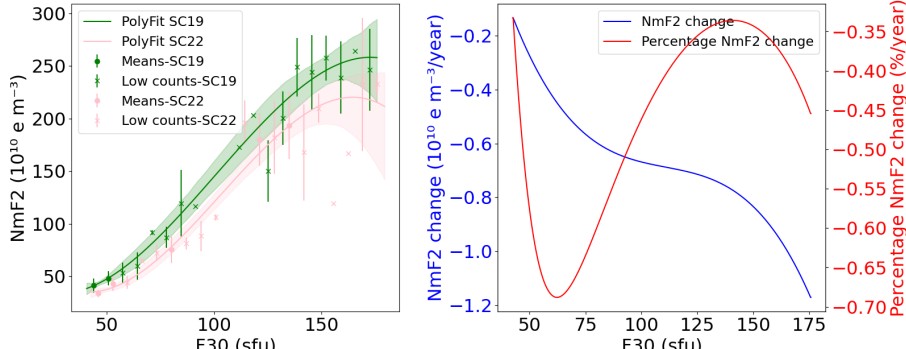

**Figure 11. Left panel**: Third-degree polynomial dependence between $NmF2$ and F30 during January at 14 LT for descending phase of solar cycles 19 and 22 with their confidence intervals (CI) indicated as shades of the same regression line color. Mean values of the bins (scatter points) and mean values with less than 10 counts in the bin (crosses) with their standard deviation (error bar for each point); **Right panel**: Absolute and percentage differences per descending year between the third-degree polynomial fitting corresponding to the descending phase of solar cycles 19 and 22.

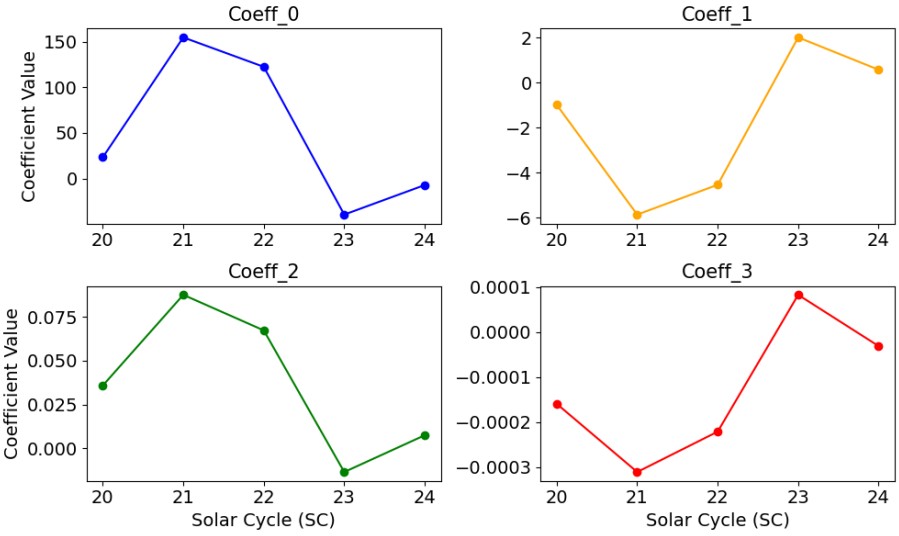

**Figure 12.** Coefficient values of the third-degree polynomial fitting between $NmF2$ and F30 during January at 14 LT from solar cycles 20 to 24

The analysis was performed for six solar cycles, including a separation of the ascending and descending phases. Based on the analysis, the following main results were obtained:





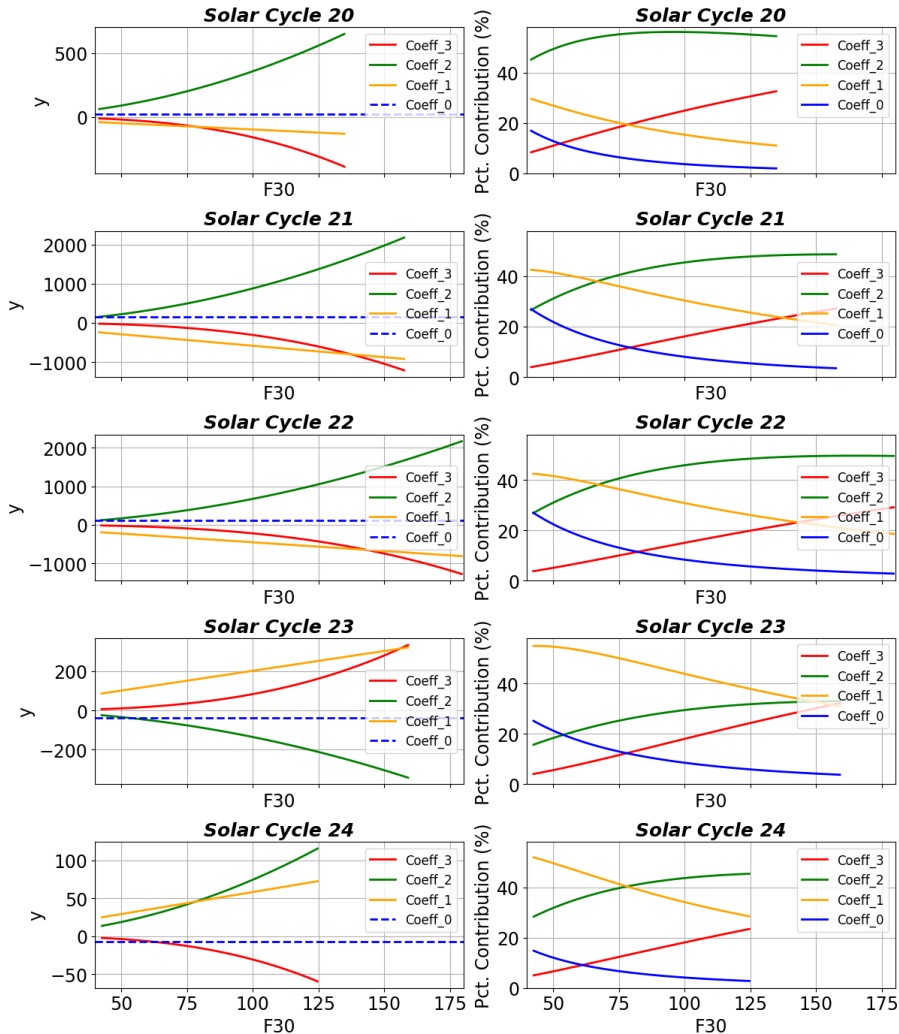

**Figure 13. Left column**: Constant (blue line), linear (orange line), quadratic (green line) and cubic (red cubic) contribution in the third-degree polynomial dependence between $NmF2$ and F30 during January at 14 LT for each solar cycle separately; **Right column**: percentage contribution from the constant, linear, quadratic and cubic component in the third-degree polynomial fitting for each solar cycle.

– The ionospheric saturation feature is visible in our $NmF2$ data. For this reason, the ionospheric response is better represented with a quadratic regression over other methods. This effect begins at lower F30 values in the ascending phase than in the descending phase.

– F30 shows the highest squared correlation value for describing the hourly $NmF2$ dependence on solar flux over time in Juliusruh in comparison with F10.7 and MgII.





- In January (a winter month), there is the highest correlation between solar flux and $NmF2$ during noon conditions, that is explained by the winter anomaly.

- The modeling of the $NmF2$ response to solar activity for each SC separately revealed a steady decrease of $NmF2$. A significant discovery is that the long-term variation is influenced by the intensity of the solar activity index. On average, $NmF2$ decreases by 0.3% to 0.44% per year for low and high solar activity index levels respectively (equivalent to $2*10^9$-$5.5*10^9$ $e/m^3$ per year). The long-term decrease becomes more significant with higher solar activity. It changes by approximately $1*10^{10}$ $e/m^3$ (3.2%) per solar cycle for small F30, and $6*10^{10}$ $e/m^3$ (4.8%) per solar cycle for F30 = 120 sfu.

This study shows that the previously reported long-term decrease of $NmF2$ at winter noon conditions at the mid-latitude station Juliusruh is reflected in the parametrisation of the $NmF2$ response to the solar activity index F30. This parametrisation method is a valuable tool to quantify long-term change in a meaningful way.

So far, the data for one ionosonde station for six solar cycles has been analysed. In order to complete the knowledge about long-term changes in the ionosphere, the analysis needs to be extended. Further studies are suggested to perform the same analysis with a greater number of ionosode stations from different parts of the world. This would provide a more comprehensive understanding of the responses across different latitudes and longitudes and help to determine if the results found are consistent.

*Data availability.* F10.7 and F30 data were taken from https://spaceweather.cls.fr/services/radioflux/. A daily resolution measurement of F10.7 is available since February 1947 and since November 1951 for F30. MgII data was taken from https://www.iup.uni-bremen.de/UVSAT/ Datasets/mgii with a daily resolution starting in 1978 (Composite MgII Index) that has been extended back to 1947 using F30 and F10.7 time-series. The Kp index data used in this work was obtained from GeoForschungsZentrum (GFZ) in Potsdam, Germany (https://kp.gfz-potsdam. de/daten). Juliusruh foF2 data can be obtained from the World Data Centre web pages at the Australian Space Weather Forecasting Centre (https://www.sws.bom.gov.au/World_Data_Centre/1/3.)

## Appendix A: Long-term changes in the dependence of $foF2$ on solar flux at Juliusruh

In this section of the appendix are shown the same results as in the paper but using $foF2$ instead of $Nmf3$, The different subsection concordance with the name of subsection in the body of the paper.

### A1 Results: Seasonal Analysis with different solar EUV proxies

Similar analysis done in Figure 4 is done in Figure A1 for the case of $foF2$.

### A2 Results: Long-term changes

Similar analysis done in Figure 5 and in Figure 6 are done in Figure A2 and in Figure A3 for the case of $foF2$.



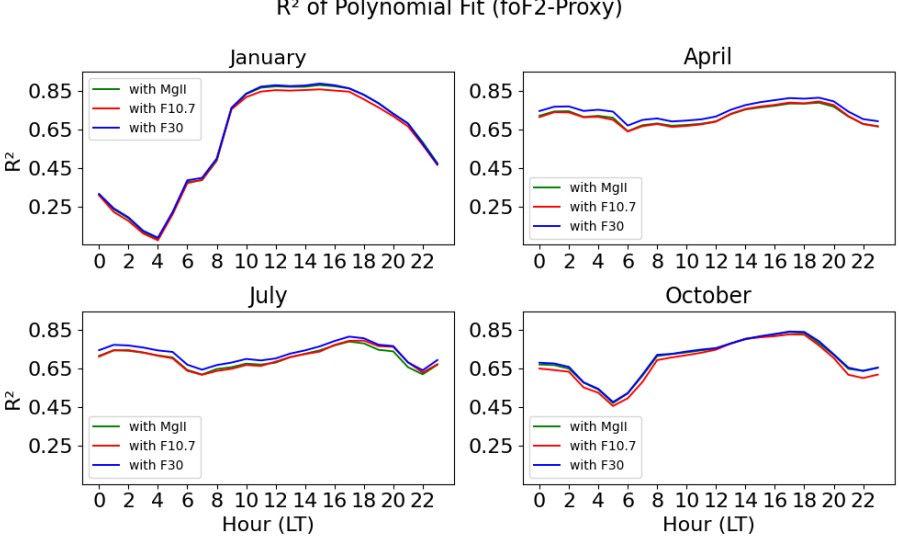

**Figure A1.** Hourly $R^2$ value of the third-degree polynomial dependence between $foF2$ and solar activity proxies: F30 (blue line), F10.7 (red line) and MgII (green line); a)in January; b)in April; c)in July and d)in October from 1957 to 2023.

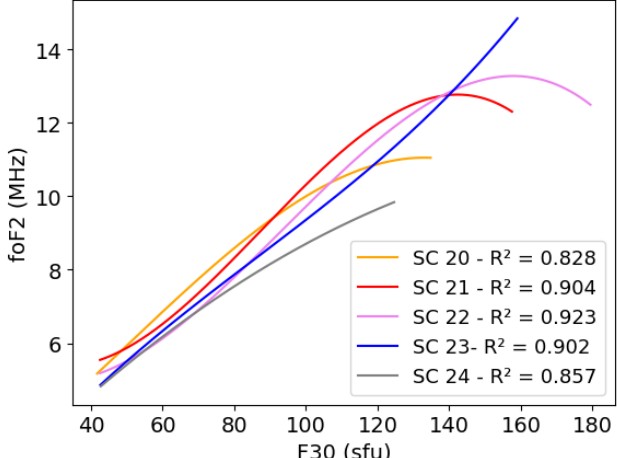

**Figure A2.** Third-degree polynomial dependence between $foF2$ and F30 during January at 14 LT for different solar cycles.

## A3  Ascending/descending phases of the solar cycle

Similar analysis done in Seccion 3.2.1 is done here for the third-degree polynomial dependence between $foF2$ and F30 of the ascending and descending phases of solar cycles.





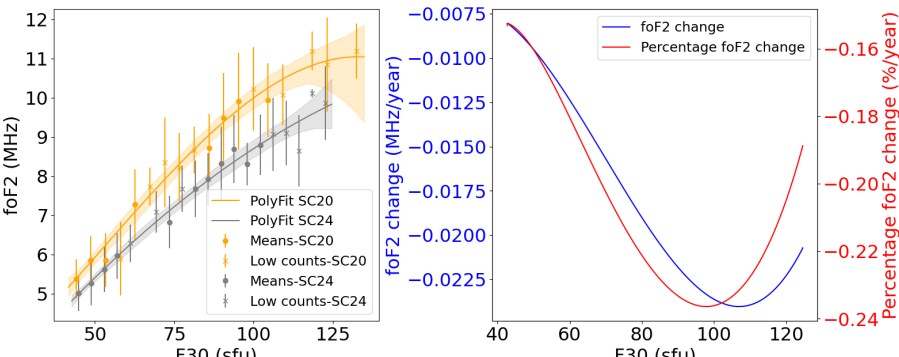

**Figure A3. Left panel**: Third-degree polynomial dependence between $foF2$ and F30 during January at 14 LT for solar cycles 20 and 24 with their confidence intervals (CI) indicated as shades of the same regression line color. Mean values of the bins (scatter points) and mean values with less than 10 counts in the bin (crosses) with their standard deviation (error bar for each point); **Right panel**: Absolute and percentage per year differences between the third-degree polynomial fitting corresponding to solar cycles 20 and 24.

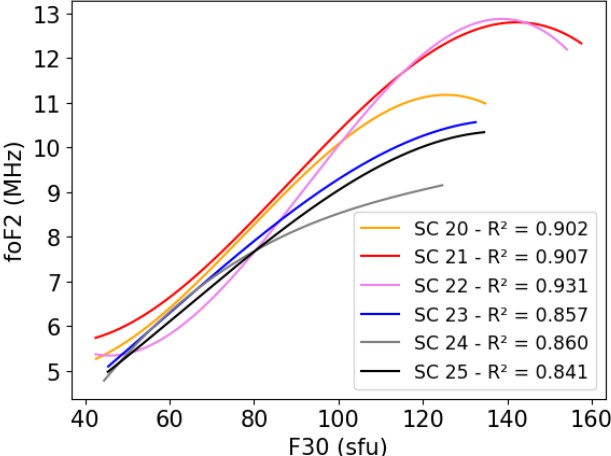

**Figure A4.** Third-degree polynomial dependence between $foF2$ and F30 during January at 14 LT for the different ascending part of each solar cycles.

## Appendix B:  Linear dependence of $NmF2$ on Solar Flux

In this section of the appendix are shown the same results as in the paper but using linear regression for the representation of the ionospheric response to solar flux instead of the quadratic fitting, The different subsection concordance with the name of subsection in the body of the paper.





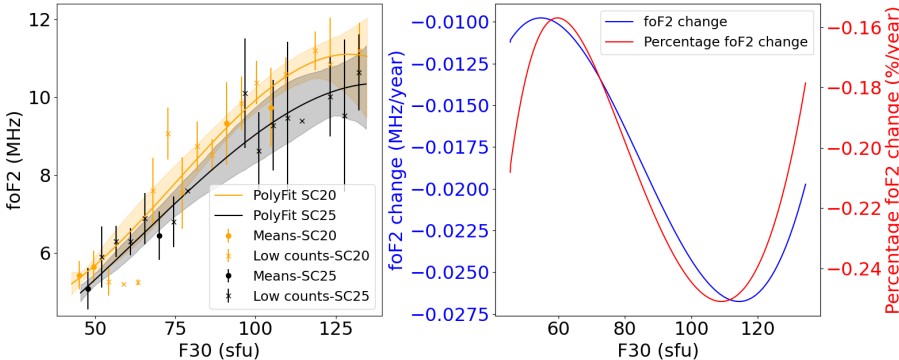

**Figure A5. Left panel**: Third-degree polynomial dependence between $foF2$ and F30 during January at 14 LT for ascending part of solar cycles 20 and 25 with their confidence intervals (CI) indicated as shades of the same regression line color. Mean values of the bins (scatter points) and mean values with less than 10 counts in the bin (crosses) with their standard deviation (error bar for each point); **Right panel**: Absolute and percentage differences per ascending year between the third-degree polynomial fitting corresponding to ascending phase of solar cycles 20 and 25.

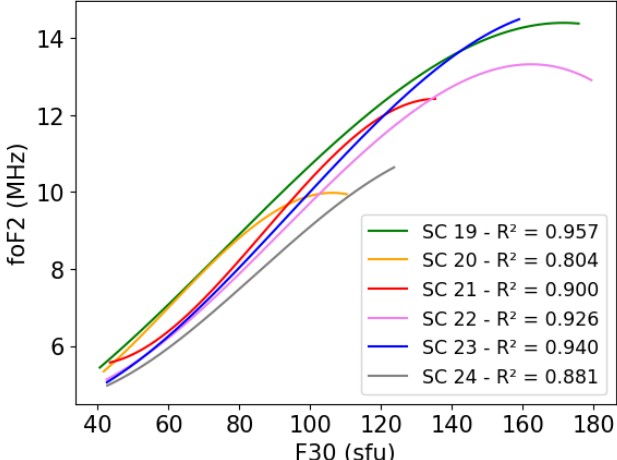

**Figure A6.** Third-degree polynomial dependence between $foF2$ and F30 during January at 14 LT for the different descending part of each solar cycles.





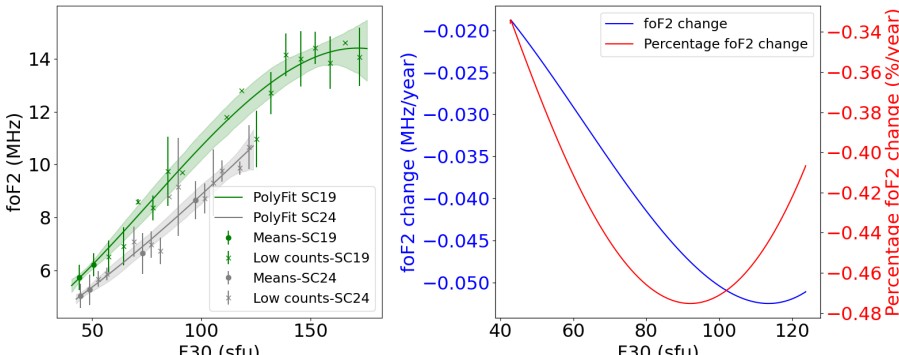

**Figure A7. Left panel**: Third-degree polynomial dependence between $foF2$ and F30 during January at 14 LT for descending phase of solar cycles 19 and 24 with their confidence intervals (CI) indicated as shades of the same regression line color. Mean values of the bins (scatter points) and mean values with less than 10 counts in the bin (crosses) with their standard deviation (error bar for each point); **Right panel**: Absolute and percentage differences per descending year between the third-degree polynomial fitting corresponding to the descending phase of solar cycles 19 and 24.

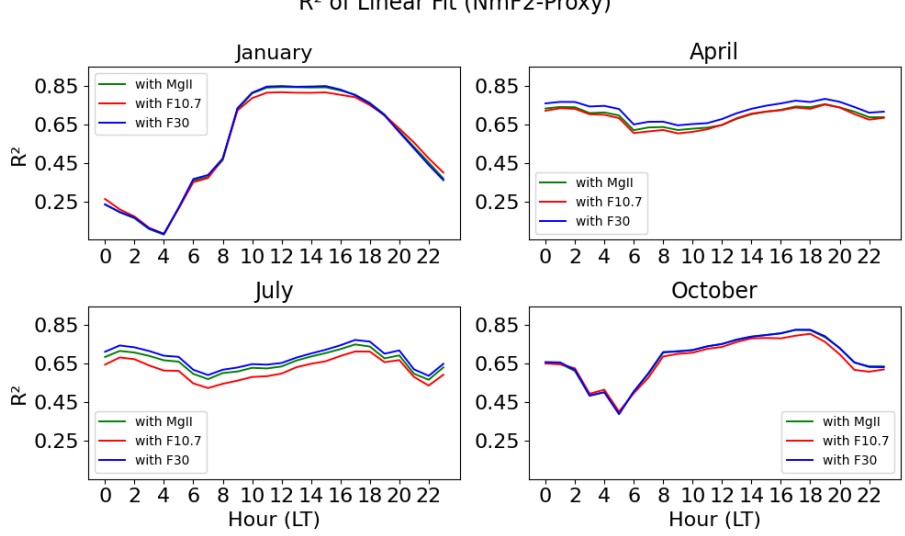

**Figure B1.** Hourly $R^2$ value of the linear dependence between $NmF2$ and solar activity proxies: F30 (blue line), F10.7 (red line) and MgII (green line); a)in January; b)in April; c)in July and d)in October from 1957 to 2023.

## B1 Results: Seasonal Analysis with different solar EUV proxies

## B2 Results: Long-term changes

*Author contributions.* MGTJR, CB, and HL conducted the conceptualization and developed the methodology. MGTJR analyzed the data, created the visualizations, and drafted the manuscript. CB and HL provided scientific support, reviewed, and edited the manuscript. JM reviewed, and edited the manuscript.



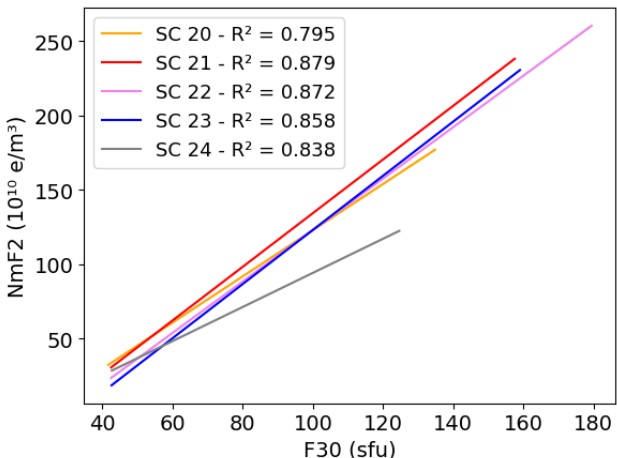

**Figure B2.** Linear dependence between $NmF2$ and F30 during January at 14 LT for different solar cycles.

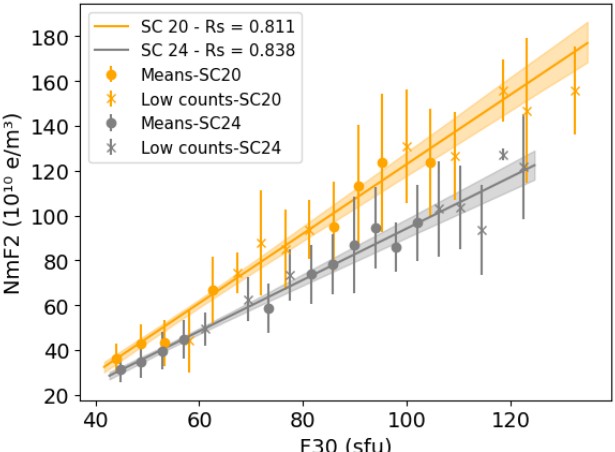

**Figure B3.** Linear dependence between $NmF2$ and F30 during January at 14 LT for solar cycles 20 and 24. Confidence intervals (CI) of the polynomial fittings are indicated as shades of the same regression line color. Mean values of the bins (scatter points) and mean values with less than 10 counts in the bin (cruises) with their standard deviation (error bar for each point).

*Competing interests.*  CB is a topical editor at Annales Geophysicae.

*Financial support.*  This research has been supported by DLR/DAAD Research Fellowships – Doctoral Studies in Germany (57622551)





*Acknowledgements.* We are grateful to Norbert Jakowski for his valuable recommendations on the ionospheric parameters used in this study, and for generously sharing his expertise with us. We also wish to thank Jens Mielich for providing access to the Juliusruh ionosonde measurements and for sharing information about it.





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
