# Peer review of "Long-term changes in the dependence of NmF2 on solar flux at Juliusruh"

_Annales Geophysicae, 2024_

## Author Response (AR1)

*The work studies the long-term change in the ionosphere at Juliusruh station by parametrization of the ionospheric response to solar activity. The variations in NmF2 with solar EUV proxies are analyzed for hourly values of each month from 1957 to 2023.*

*A brief description of each solar proxy used in the study is given in subsection 2.1. A special attention is paid to getting rid of wrong data in NmF2 related to human errors or to geomagnetic disturbances. The authors call the process as "cleaning". The cleaning consisted of two steps. In the first step, the values "that fall far outside the natural range of NmF2" were deleted. In the second step, the effects of geomagnetically disturbed days were withdrawn. The results of the cleaning are shown in Table 2. It shows that the number of points left for the analysis is 65.5% of the initial number of points.*

*The authors compare the results of using three solar proxies (F30, Ly-a and MgII) and conclude that F30 is the best. That agrees with the opinion of many researchers.*

*The following method was used for the analysis. The hourly NmF2 values were presented as a third-degree polynomial of the corresponding solar proxy. The quality of that presentation was estimated by the correlation coefficient squared which actually is the determination coefficient of the Fisher F-test. Figure 2 shows the dependence of NmF2 on F30 for particular conditions (14 LT, January) in solar cycle 22. It shows a linear and polynomial approximation of the data and the deviations of the points with a small (less than 10) number of points in the bin.*

*Analysis of the seasonal effects shows (subsection 3.1) that there is a well pronounced dependence of the $R^2$ value on LT in winter months (January is shown as an illustration), but there is no such dependence in equinox and winter months. That exactly agrees with the results obtained by the analysis of the foF2 data at 11 stations of the Northern and Southern hemispheres. (Geomagnetism and Aeronomy, 2024, Vol. 64, No. 2, pp. 224–234).*

*In subsection 3.2, the changes in the NmF2 dependence on solar proxies in various SA cycles are analyzed. Fig. 5 shows that there is a clear tendency in the curves for later cycles to go below the curves for early cycles. In my opinion, the most interesting is the left panel of Fig. 6 that demonstrates that for the same value of the solar proxy, the NmF2 values for cycle 24 are lower than for cycle 20.*

*The authors analyze the so-called "hysteresis effect". First, they show that the effect does exist even if the geomagnetically disturbed days are withdrawn from the analysis. But the authors conclude that it does not influence substantially the final results.*

*Figures of the Fig. 6 type make it possible to derive trends in NmF2 (foF2). The authors compare their results with some estimates of the trends in NmF2 and foF2 published by other groups and conclude that there is a reasonable agreement.*

*Important conclusion of the paper is that the trends in NmF2 (foF2) depend on the SA level. The trends are the lowest at low activity and the highest at high one.*

*I consider the paper as an important contribution to the studies (currently very popular) of the long-term changes in the dependence of ionospheric parameters on SA and the corresponding trends in these parameters first of all, in NmF2 (foF2). I recommend publishing the paper with minor revision. My comments are the following.*

1. *"Figure 2. Juliusruh ionosonde hourly observations of NmF2". What is shown actually? If the monthly mean values, then for which month? Do the data present daily averaging?*

- The data presented in Figure 2 is the full experimental dataset, which includes the NmF2 data derived from the foF2 experimental data in Juliusruh with hourly resolution for all months from 1957 to 2023.

2. *In my opinion, the paper is overloaded by figures. I recommend withdrawing Figs 12 and 13. It could be done without a serious revision of the text.*

- We believe it is better to directly delete Figures 12 and 13 from the manuscript, with corresponding modifications to the text.

**Anonymous Referee #2**

*Comments on the paper Long-term changes in the dependence of NmF2 on solar flux at Juliusruh*

*This paper studies the trend of NmF2 calculated from measurements at Juliusruh station during the period 1957 to 2023. To study this trend, a regression analysis is performed using mainly F30 as a solar proxy for a cubic fit in the equation. Geomagnetic activity (measured days with kp values<3) and also some "natural" points as outliers are filtered.*

*For the study, hourly values were analyzed for all days of the mentioned period and then separated by seasons: January (Winter), April (Spring), July (Summer) and October (Autumn). I have my most profound observations about this methodology. Precisely for the analysis of the trend, daily values of the hours are being used and not the monthly median as reported in all the literature in trend studies (Laštovička mentions precisely its use as a way to eliminate geomagnetic effects. Laštovička, J.: Dependence of long-term trends in foF2 at middle latitudes on different solar activity proxies, Advances in Space Research, 73, 685–689, 2024.). Although it is later filtered by magnetic activity (those days with kp greater than 3) the final results in trend are quite higher than those obtained with other methods by other authors for the same season. I do not see the contribution of doing it this way. The study should be repeated but taking the monthly median of the hours and comparing both situations. Another option is to apply the same method in other seasons as this same work suggests in its conclusions.*

- In previous studies, monthly median values were used to remove geomagnetic influences. However, in our hourly data, we address this problem by excluding days with a Kp index greater than 3. Despite the method being different from previous approaches, the trend we obtained is consistent with recent findings for the Juliusruh station, as stated in lines 257-263 of this manuscript, and we are considering an ionosphere without geomagnetic perturbation as well.

   Repeating this study using monthly medians instead of daily values will reduce the accuracy because instead of having approximately 160 points per month (January) at a particular LT (14LT)

in a solar cycle, we will have only 11 values per each solar cycle (one per year for each January 14LT).

In the new draft, a seasonal analysis was added (see line 267 in new draft), however, the interpretation must be done carefully. Our seasonal analysis showed similar mean trends. This is good. We prefer avoiding additional figures because we have already many and a tabular overview might be sufficient.

*On the other hand, the regression is done using cubic terms, and in principle it is mentioned that they are not negligible when calculating because they supposedly contribute a large percentage of the total as indicated in figure 13. If we look at the final results of the trends obtained by this work we see that we reach values of 3 to 4.4% decadal (for 90 sfu) which means that in approximately 230 years the value of NmF2 will be zero!!! (for 120 sfu it is even more significant) Precisely this method gives trend values greater than those reported in other works for that same station (line 261 says that Cnossen is -8kHz in the case of foF2 per year being that the lowest value of the change interval found in this work, line 257). It would be good to compare with the trend results obtained from the linear regression analysis, which should be similar to that of other authors except that here the complete values of all hours are being used, not the monthly medians. On line 162 it says that R2 calculated with linear regression does not differ significantly from the cubic one, which is not what they are trying to show.*

- There are not many papers with data on NmF2 trends, but to compare our results with other publications, we used the same method for the foF2 parameter. Our findings show that our decrease aligns with the results from other studies. For instance, we observed a decadal reduction of 1.5-2.3% (or 8-24 kHz/year) for foF2 (1964-2014). This is consistent with the reported decadal trend of -1.8% in Table 2 of Laštovicka (2024) for Juliusruh (1976-2014), as well as the trend of -7.7 kHz per year for Juliusruh (1959-2005) as reported by Cnossen and Franzke (2014). Although our results align with previous findings, it is important to note that a direct comparison is challenging due to the different periods analyzed in the various studies.

*In this work, an analysis was made using the bootstrap method and when we look at figure 3, I see that the high F30 values (precisely those of saturation) have less weight than the rest. The last values are all with cruises. Perhaps a different statistical analysis such as Akaike Information Criterion (AIC) or Bayesian Information Criterion (BIC) helps to identify the contribution of each term in the adjustment and can be used to choose the appropriate number of terms.*

- *The bootstrap method was used in this work to compute the confidence interval of the fittings. For that reason, when the uncertainty is higher, it generates a bigger confidence interval as is visible in Figure 3. It is suggested to use the Akaike Information Criterion (AIC) and/or Bayesian Information Criterion (BIC) instead of the Bootstrap method. AIC and BIC do not compute confidence intervals as Bootstrap does. However, these criteria can reinforce the selection of a cubic fitting, which seems to be the purpose of this comment. Therefore, we computed and compared AIC and BIC for linear, quadratic, and cubic regression models. Finally, we found the lowest AIC and BIC values using the cubic fitting for most solar cycles. This supports the cubic fitting as the best of these three models. That result is mentioned in the discussion section of the new draft (line 250).*

|          | SC20          | SC21          | SC22          | SC23          | SC24          |
|          | AIC           | AIC           | AIC           | AIC           | AIC           |
|----------|---------------|---------------|---------------|---------------|---------------|
| Linear   | 1284.808751   | 1276.974256   | 1170.981588   | 1626.918080   | **1511.385254** |
| Quadratic | 1284.638496  | 1278.895970   | 1171.705581   | 1571.238455   | 1513.384188   |
| Cubic    | **1283.949986** | **1252.367396** | **1137.730541** | **1570.338215** | 1515.179379 |

*In intense cycles such as SC20, it is assumed that the cubic adjustment being made is closer to the original curve and therefore the trend if we look at the residuals should be lower, the straight line should not have as much negative slope with respect to obtaining the trend if we use a linear or quadratic adjustment as a higher degree. However, in this work using a cubic adjustment we arrive at high trends. How do we justify this?*

- In comparison with SC21, SC22, and SC23, SC20 is not as intense. This can be observed in Figure 1. SC20 is, however, more intense than SC24. It is unclear whether you are referring to a specific plot when mentioning "residuals" and "straight line." Could you please provide further clarification?

*Perhaps the paper intends to propose the full-hour method (daily values) and cubic regression as an alternative method to the usual one. In any case, this approach is yielding quite high trend values, which leads one to think that we will soon run out of ionosphere. That is why I am not so sure if the method can help these studies, even, as the paper says, the extension to other stations can give us clues as to whether this type of analysis again gives higher values than those calculated in a traditional way.*

- *The concept of trends in this paper differs from the traditional approach. We analyzed the variation in the ionospheric response using solar cycle periods. While the idea of linearity in the trend still exists, this method allows us to differentiate responses to high and low solar activity. Our results align with previous reports, as anticipated. This idea was emphasized in the last paragraph of the discussion section of the new draft.*

*In any case, I consider that the paper can be published, making the considered revisions and the suggested corrections.*

*Minor corrections*

*Line 14 I think the years from 1954 to 2019 are wrong*

- The correct period is from 1964 to 2019, which includes the beginning of SC20 and ends with SC24.

*Line 45 Ma's 2009 paper does not work with third degree polynomials but with second degree*

- *That is correct. It was modified in the new draft.*

*Line 63 says appendix 5.5 but it does not exist*

- Typing error. It must say "Appendix A". *It was modified in the new draft.*

Line 66 says "coorelation" and it should be correlation

- *That is correct. It was modified in the new draft.*

Line 78 a period is missing at the end

- *That is correct. A full stop is missing. It was modified in the new draft.*

Line 86 Wouldn't that be 7 solar cycles?

- *The period (1957-2023) includes SC19 to SC25, but in this period SC19 and SC25 are not complete, we are just considering the descending phase of the first one and the ascending phase of the last one. Maybe a clear correction must be "five complete solar cycles" to avoid confusion. It was modified in the new draft.*

Paragraph 100 would be Kp equal to or less than 3, the equal is missing based on what line 109 says

- *That is correct. It was modified in the new draft.*

Line 167 if we have data from 1957 to 2023 that is from cycle 19 to 25 and not from 20 to 24

- There are no mistakes in this sentence: "To achieve this, we divide the period from 1957 to 2023 into different solar cycles based on Table 1 and consider the period between solar cycles 20 to 24 when the observations are available for each full solar cycle." Because it means that we only consider the data corresponding to SC20-SC24 from the entire period.

I don't understand line 187 which says that from cycle 20 to 24 it varies between 3.2 to 4.8 per decade. Who is this referring to? It is almost 5% in 200 years we are at 0. In the abstract it says 16% to 24% which is for 90 sfu and in this sentence it says it would be 3.2 to 4.8. There are inconsistencies in the results. It may refer to 120 sfu as it says in line 313. Please clarify this

- In line 186 appears a variation of 0.3-0.44% per year, which is 3-4.4% in a decade. 3.2-4.8% corresponds to a variation per solar cycle between SC20 and SC24. The variation between SC20-SC24, 5 solar cycles (approx. 55 years), is of 16-24%. However, after this comment, we realized that the trend, by definition, does not imply solar cycle variation. Therefore, showing our results using the variation per solar cycle could have confusing implications. In the new draft, we maintain the trend results only in terms of year and decade to avoid confusion.

Line 228 mentions the Liu and Chen 2009 paper to justify that the cubic fit is fine but in that paper in the conclusions it says that they recommend quadratic fit for TEC because higher orders are not significant!! Please clarify.

- *That is correct. Badly reference. It was modified in the new draft.*

Line 233 a period is missing at the end of the sentence

- That is correct. A full stop is missing. *It was modified in the new draft.*

*Line 236 in 5? I don't understand*

- It must say "in Figure 5". *It was modified in the new draft.*

*Figure 13 should say (red line) but it says (red cubic)*

- That is correct. *The plot was removed.*

*Line 328 would be NmF2 and not f3 as it says*

- *That is correct. It was modified in the new draft.*

*Line 339 says quadratic but it would be cubic, important result of this work!*

- *That is correct. It was modified in the new draft*

*Figure 12 and 13 talk about "coeff" when in reality the equation is a0, a1, etc.*

- *That is correct. However, now in the new draft Figures 12 and 13 were deleted from the draft.*

*Line 317 says six solar cycles but I think there are 5, right?*

- Same discussion as in the previous point of Line 86. *It was modified in the new draft.*

**Editor: Jan Laštovička**

*This paper has potential to become valuable contribution to the special issue on long-term trends. However, first it must be re-reviewed by reviewer #2.*

*My comment to authors. Kp equal to or smaller than 3 does not guarantee the removal of geomagnetic storm effects during periods of deep solar cycle minima (e.g. Buresova et al., Adv. Space Res., 54, 185-196, doi: 10.1016/j.asr.2014.04.007*

- To answer this question, we made a further analysis of our data. We decided to filter using Kp>=2 and 48hs after it occurs in one year around each solar minimum. In our paper, Table 2 includes the quantified analysis of the NmF2 data for the cleaning method used. Now, if we filter kp>=2 and 48hs after its occurrences for one year around each solar minimum, we will filter 5% more of the total data. It means that we are ignoring 62% of the period corresponding to the deep solar minimum condition.

  The work of Buresova et al. found that the positive effect of the magnetic storm in NmF2 has a long-extended period of storm-induced changes after the storm. Therefore, we also try to filter after 72 hs of a kp>=2 occurs. In this case, we filtered 5.6% more of the total data, which means approximately 66% of the deep solar minimum period.

The following plots for some solar cycles correspond to the last case (kp2 and 72hs after), which is the "worst" case, due to the higher amount of data filtered. "PolyFit SC…" represents the fitting with the data we used in the paper (just filtering kp>=3) and "PolyFit SC…_min" corresponds to the fittings for each solar cycle after filtering the periods of solar minima. As is visible here, there are no significant differences between the fitting models and the original cleaning method. For that reason, to simplify the cleaning method and to use more amount of data for our analysis, we believe is better to keep the analysis as it is. This point is explained in the new version of the draft (line 224).